# A Systematic Compilation of the Problems Encountered by Teachers and Students in Science and Arts Centers in Turkey

Harun Şahin [1] , Süleyman Karataş [1,*] , Muhammed Ali Özkan [2] , Ramazan Gök [1] , Meriç Eraslan [3] , Erdoğan Köse [1] and Nilgün Azeken [4]

1 Department of Educational Sciences, Akdeniz University, Antalya 07000, Turkey
2 Ministry of National Education, Muş 49100, Turkey
3 Department of Sport Science, Akdeniz University, Antalya 07000, Turkey
4 Ministry of National Education, Antalya 07000, Turkey
* Correspondence: skaratas@akdeniz.edu.tr; Tel.: +90-5055009080

**Abstract:** In Turkey, Science and Arts Centers (BİLSEM) were opened and put into practice in 1995 in order to support gifted and talented students who are different from their peers in academic, social, and artistic fields in addition to their formal education. However, as in every educational institution, many problems have emerged in BİLSEMs, which teachers and students try to overcome. In this research, we aim to address the problems that teachers and students in BİLSEMs face or experience. To this end, the study employed a systematic compilation method, a qualitative research method, and theses, articles, papers, and books that were published between 1995 and 2022 and could be accessed were included in the study. In accordance with the purpose of the study, the research data were compiled by entering the keywords "BİLSEM", "Gifted", and "Talented" in the databases and analyzed with content analysis. As a result of the reviews, 46 articles, 15 theses, 1 paper, and 1 book focusing on teacher and student problems were included in the study. In light of the results obtained in the study, it was concluded that the problems experienced by teachers include shortages in materials, physical conditions and infrastructure, insufficient professional development opportunities, the programs not being functional and adequate, and student absenteeism and that the problems experienced by students include the intensity of the program and participating in too many activities, insufficient infrastructure and a lack of materials, being excluded from their circle of friends, and high expectations.

**Keywords:** science and arts center; BİLSEM problems; gifted; talented; review





## 1. Introduction

Culture is an important element for the survival of societies. In order to survive, societies deliberately and consciously attach importance to cultural transfer and aim to train people to adapt to their cultural contexts. Undoubtedly, education is one of the most important aspects to train humans for society's cultural structure [1]. In fact, education systems aim to train individuals who know themselves and their culture, are aware of their own potential, and can develop their potential. The education system and education programs in Turkey have adopted a student-centered education approach. Educational processes where the student is placed at the center aim to uncover individual differences and the potential of individuals. In an educational system where the difference of each student is brought to the fore, the education process of individuals who are considered to be gifted gains greater importance. The main determining factor for the achievement of sustainable development is education [2]. Sustainable development education emphasizes education that will help students to create and develop a more sustainable economy, environment, and social order, think about the future, and make strategic plans [3]. Due to the existing potential of gifted students, they are of great importance in the sustainable development of countries.

Gifted individuals are defined in the literature as "individuals who have higher cognitive performance than their peers and have high potential in a specific talent" [4–6]. In the Ministry of National Education Directive on Science and Arts Centers (2007), gifted students are defined as "individuals determined by experts to have a potential beyond their age in areas such as arts, leadership, production and creativity". In addition, it has been stated that gifted children show a development that is ahead of their peers in the physical domain, and they speak and walk earlier [7]. The common point of these definitions is that gifted individuals have different and superior characteristics and abilities from their peers. Thus, it is obvious that, in educational systems, individual differences are important, and these differences are of particular importance in the education process of gifted individuals. Taking into account the needs of gifted students and making plans in line with their interests and needs will contribute not only to the society in which they live but also to humanity. Gifted individuals are individuals who, in addition to possessing theoretical knowledge, are inquisitive, questioning, creative, have problem-solving skills, can take firm steps towards self-realization, and will contribute to the production of knowledge [8]. In the research conducted by Schack and Starko, referring to the opinions of teachers from different branches ($n = 308$), it was found that there is a parallelism between being good at lessons and being gifted, and Neumeister, Adams, Pierce, Cassady, and Dixon [9], regarding teachers' opinions on gifted individuals, defined individuals as independent, highly motivated, and creative, learning faster than their peers and having a high understanding capacity. In the study conducted by Moon and Brighton [10], the majority of teachers ($n = 434$) had traditional views when describing gifted individuals. In the study of Gökdere and Ayvacı [11], teachers stated that they do not have enough information to define gifted individuals. Akar and Akar [12] supported the findings of Gökdere and Ayvacı [9] in their study with 155 teachers. When the findings obtained in the research are evaluated, and when the opinions of the teachers on the definition of gifted students are considered, it can be seen that concepts such as course success, creativity, and motivation come to the fore, and as in the research findings of Gökdere and Ayvacı [11], it can be seen that some teachers do not have sufficient knowledge about the subject. Therefore, teachers working with gifted individuals should be equipped and have sufficient knowledge in this regard.

Indeed, it is evident that traditional classroom environments are insufficient to attend to the differences of gifted individuals and support their development. It has been stated that, if the education of these individuals is limited to traditional classrooms, their abilities may be dampened and they may be unhappy [13]. Renzulli [14] developed the Three Ring Theory. Renzulli argues that gifted individuals have above-average motivation and creativity. According to him, superior talent emerges with the interaction of the three rings: giftedness, motivation, and creativity. The absence of being above average in any of these circles means the absence of being gifted.

According to Renzulli [15], a high IQ is not sufficient to be gifted. In addition, the gifted student must have motivation and creativity. According to Renzulli [16,17], the prevalence of gifted individuals in society is between 2% and 3%, while the general or special talent circle is between 15% and 20%. As can be understood from this expression, giftedness, which is expressed as general and special abilities, is not limited to a certain degree and is stated as being above average.

An education process in which gifted students are separated from their peers has been considered detrimental to a democratic education process [18,19]. However, a privileged education has been considered a necessity for gifted students and the potential of these individuals will be increased by the education offered to them in accordance with their needs [8,20–22]. The general opinion that emerges regarding the education of the gifted is that the normal school curriculum is insufficient for the education of these children. For this reason, various training programs on the subject have been established in different countries. There is no special school for gifted children in the UK. However, there are schools where they can obtain education in fields such as church grove, music, ballet, and

theater. The differentiation of education in schools in England according to student abilities has created the opinion that there is no need for a separate education program for gifted students [23]. In Germany, an integrative education is planned for gifted students and special classroom practices are carried out when necessary. In Finland, important steps have been taken in special education, but necessary studies are not carried out for gifted individuals [24]. In Turkey, Science and Arts Centers were opened to meet the privileged education needs of gifted students.

Science and Arts Centers (BİLSEMs) offer an after-school program in line with "The Decree Law No. 573 on Special Education published in the Official Gazette dated 6 June 1997 and numbered 23011 and the Special Education Services Regulation published in the Official Gazette dated 31 May 2006 and numbered 26184, (2013)", and the first Science and Arts Center was opened in 1995. Students can attend BİLSEM courses either in the evenings on weekdays or during the weekend. BİLSEM provides education opportunities for gifted children in primary, middle, and high school in order to develop their potential. The BİLSEM education model adopts a project-based approach [25–27]. The selection of students for BİLSEM takes place in three stages. The teacher at the school attended by the student nominates the student; the nominated student is subjected to the group intelligence test; and if he/she passes it, the selection process is completed with the individual intelligence test. If the total score of the student is higher than 130 IQ points, the student is entitled to enroll in BİLSEM. The enrolled student completes his/her education in a context that adapts, supports, and recognizes individual talents, developing special talents and project programs. In the BİLSEM education model, while students continue the education process with their peers during formal school hours, they receive the BİLSEM program education outside of formal school hours. The training areas and framework program of BİLSEM are pre-determined, while the application process and methods are left to the discretion of the implementer [28]. This results in different applications. Therefore, it is not possible to achieve unity in BİLSEMs in terms of educational processes. In addition, the uncertainty in the program may cause difficulties in relating it to the formal school curriculum. These difficulties also create different problems.

In this context, the purpose of the current study is to collect studies that deal with the problems experienced by students attending BİLSEMs and the teachers who manage the process. To this end, a literature review focused on articles published, papers presented in congresses and symposiums, and books published between 1995 and 2022 was conducted, and 46 articles, 15 theses, 1 paper, and 2 books published on the subject were included in the study.

The following are the research questions of the current study conducted to reveal the problems experienced by students and teachers in Science and Arts Centers:

1. What is the distribution of the studies focused on the problems experienced by students attending BİLSEMs and the teachers who manage the process across the years?
2. What research methods have been used in the studies focused on the problems experienced by students attending BİLSEMs and the teachers who manage the process?
3. What problems have been mentioned and how frequently in the studies focused on the problems experienced by students attending BİLSEMs and the teachers who manage the process?

## 2. Materials and Methods

### 2.1. Research Design

As we aimed to determine the studies focused on the problems experienced by the teachers and students in Science and Arts Centers and to systematically gather the data obtained from these studies in the current review, the systematic compilation method, a qualitative research approach, was used as the research design. A systematic compilation is a systematic and unbiased review of studies on the same subject, in accordance with pre-determined criteria, in order to find answers to research questions in the context of the main purpose of the research, checking the validity of the studies found and synthesizing

them [29]. A systematic compilation is a research method that produces the strongest foundations for evidence-based practices. Through this research method, the findings of multiple studies suitable for the research are compiled and the best evidence is created by making the necessary analyses [30]. According to Aslan [31], a systematic compilation is a scientific review in which studies are reviewed in detail and their findings are synthesized. The common point of the definitions is that systematic compilation provides a comprehensive research opportunity and that the studies are included in the study in the context of the pre-determined criteria, with the studies being determined and their findings synthesized.

### 2.2. Data Collection and Analysis

Studies published in national journals, theses retrieved from the Higher Education Thesis Centre, papers presented at congresses and symposiums, and other documents on the problems experienced by teachers and students in BİLSEMs between 1995 and 2022 were included in the current study. However, since there was no study on the subject between 1995 and 2007, the scope of the study was narrowed to cover the period between 2007 and 2022. In accordance with the purpose of the study, the research data were obtained by entering the keywords "BİLSEM", "Gifted" and "Talented" in the databases of ULAKBİM. In content analysis, similar data are gathered around the axis of certain concepts and themes, and then they are interpreted by arranging them in a format that can be understood by the readers [32]. The literature review continued until September 2022. As a result of the review, 46 articles, 15 theses, 1 paper, and 2 books focused on the problems experienced by students and teachers in BİLSEMs were included in the study.

The criteria used in the inclusion of the studies in the current study were as follows: 1. the studies had to have been carried out between 1995 and 2022; 2. the studies had to have samples consisting of participants living within the borders of Turkey; 3. the study findings had to include student and teacher problems about BİLSEMs; and 4. the study's methods had to be clearly defined.

### 3. Results

In this section, the names of the studies retrieved, their methods, and tables of the problems separately experienced by students and teachers are presented.

Studies carried out from 1995, when Science and Arts Centers were first established in Turkey, to September 2022 were examined and a list of the names, types, authors, and years of the studies involving teacher and student problems is shown in Table A1 of Appendix A.

According to Table A1 of Appendix A, there were 46 articles, 15 theses, 1 paper, and 2 books, a total of 64 studies, included in the current study. Studies were coded as S1, S2, S3, etc.

The distribution of the above-mentioned studies across the years is shown in Table 1.

When the distribution of the studies shown in Table 1 is examined, it can be seen that the studies were produced between 2007 and 2022. The highest number of studies seems to have been published in 2019 (*f* = 13) and there is a trend of increase in studies in recent years.

In Table A2 of Appendix B, information about the research methods employed in the studies is presented. The names of the studies are not shown here; rather, the codes shown in Table A1 of Appendix A are used.

As can be seen in Table A2 of Appendix B, case studies are the most preferred research model, BİLSEM teachers are the most preferred participants, the semi-structured interview form is the most preferred data collection tool, and content analysis is the most preferred method of analysis.

Problems experienced by students and teachers in BİLSEMs were listed separately for students and teachers, and the problems that emerged were grouped into categories. The findings involving student problems are presented in Table 2, and the findings involving teacher problems are presented in Table 3.

**Table 1.** Distribution of the studies across the years.

| Year | n |
|---|---|
| 2022 | 2 |
| 2021 | 10 |
| 2020 | 9 |
| 2019 | 13 |
| 2018 | 8 |
| 2017 | 4 |
| 2016 | 3 |
| 2014 | 6 |
| 2013 | 3 |
| 2012 | 2 |
| 2011 | 2 |
| 2010 | 1 |
| 2007 | 1 |

**Table 2.** Student problems in BİLSEMs.

| Category | Problems | f |
|---|---|---|
| Arising from themselves or their environment | Being excluded from their circle of friends | 7 |
| | Too many expectations for them | 6 |
| | Not being well understood and communication problems | 6 |
| | Asking too many questions | 4 |
| | Insufficient individualization and boredom at school | 4 |
| | Misguidance of parents and teachers in career planning | 3 |
| | Being active and lack of attention | 3 |
| | Being made an example even though they do not want to stand out | 2 |
| | BİLSEM does not meet expectations | 2 |
| | Being perceived as someone superior at school | 2 |
| | Receiving negative reactions from the people around them when their classmates obtain higher grades than them | 1 |
| | Families not encouraging their abilities | 1 |
| Arising from the program | Insufficient training program | 5 |
| | Problems in the process of measurement and evaluation | 2 |
| | No foreign language other than English | 2 |
| | Not using special publications for BİLSEMs | 1 |
| | Overlaps between the activities they want to participate in | 1 |
| Arising from the institution | Intensity of the activities and having to participate in too many activities | 14 |
| | Insufficient infrastructure and lack of materials | 10 |
| | A low number of tenured teachers | 7 |
| | Transportation to BİLSEM | 4 |
| | Education at BİLSEM is parallel to what is conducted at school | 3 |
| | Teachers not using appropriate teaching methods | 3 |
| | Inadequate class hours allocated to some lessons and activities | 3 |
| | Absence of sports activities | 2 |
| | Inadequate opportunities offered to graduate students | 2 |
| | Problems in the methods used for diagnostic purposes | 2 |
| | Crowded classrooms | 2 |
| | Compulsory participation in local and national competitions | 1 |
| | Scientific conversations are boring and ineffective | 1 |

**Table 3.** Teacher problems in BİLSEMs.

| Category | Problems | f |
|---|---|---|
| Arising from students | Absence of students | 12 |
| | Reluctance and low motivation in students | 7 |
| | Students being tired | 4 |
| | Students' adaptation problems | 2 |
| | Students arriving late and timing problems | 2 |
| | Students not being able to express themselves | 2 |
| | Students' lack of responsibility | 2 |
| | Students demanding excessive attention | 1 |
| | Students' devaluation of BILSEM | 1 |
| | Lack of discipline on the part of the students | 1 |
| Arising from the program | The program is not functional or adequate | 15 |
| | Problems experienced in the measurement and evaluation process of the program | 8 |
| | The program is not up to date | 6 |
| | Lack of standards in the objectives of the program | 3 |
| | The program is not directed at the students' interests and needs | 3 |
| | Lack of interdisciplinary subjects in the program | 2 |
| | Lack of elements supporting psychological development in the program | 1 |
| | Insufficient activity times in the program | 1 |
| | The program is not integrated into formal education | 1 |
| Arising from the institution | Shortage of materials, inadequate physical conditions and infrastructure | 28 |
| | Insufficient professional development opportunities | 16 |
| | Inadequacies in the process of diagnosing students | 13 |
| | Too many students and crowded classrooms | 12 |
| | Working hours and time | 10 |
| | Lack of teachers, particularly tenured teachers | 9 |
| | Insufficient financial opportunities, problems with personal rights | 8 |
| | Insufficient teacher qualifications | 7 |
| | Communication problems with parents | 6 |
| | Lack of cooperation with other schools and universities | 5 |
| | Poor institutionalization | 4 |
| | Being far from the city center | 3 |
| | Lack of support by the Directorate of National Education | 2 |
| | Insufficient activity books | 2 |
| | Too many procedures | 2 |
| | Lack of support offered by administrators to teachers | 2 |
| | Working with younger age groups | 2 |
| | Short break time, program intensity | 2 |
| | Communication problems between administrators and teachers | 1 |
| | Populist approaches | 1 |
| | Difficulty in scheduling workshops | 1 |
| | Student progress is not recorded | 1 |

As can be seen from Table 2, student problems in BİLSEMs are divided into three categories: problems arising from themselves and their environment, problems arising from the program, and problems arising from the institution. In the category of problems arising from themselves and their environment ($f$ = 41), the problems of being excluded from their circle of friends ($f$ = 7), too many expectations for them ($f$ = 6), not being well understood and communication problems ($f$ = 6), asking too many questions ($f$ = 4), and insufficient individualization and boredom at school ($f$ = 4) come to the fore. Below are some sample statements from the selected studies regarding the category of problems arising from themselves and their environment.

> "Students stated that they could not find friends in their formal education schools, they were excluded and they had difficulty in communicating." (S54)

> "They look very different in their classroom environment, so they are excluded." (S10)

> "Being in more than one institution creates high expectations. This means too much stress for me." (S15)

> "Problems arise due to high expectations from students." (S22)

> "Gifted children's not being understood well and communication problems." (S22)

> "At school, there is no problem that my daughter has with the people around her, but at home she overwhelms me with her endless questions, I cannot convince. I always try to persuade her by talking to her at home. If it's sixteen hours we're awake at home, I have to talk to her for ten hours." (S56)

> "The problems of the child's being bored or not being academically challenged come to the fore." (S35)

In the category of the problems arising from the program ($f$ = 11), the problems of insufficient training programs ($f$ = 5), problems in the process of measurement and evaluation ($f$ = 2), and no foreign language other than English ($f$ = 2) come to the fore. Below are some sample statements from studies on the problems in the category of problems arising from the program.

> "Limited variety of programs offered. It is not right to apply gifted education in a group by only enhancing the objectives. It can be investigated and seen which system is efficient. Teachers must be highly qualified. Modules should be created and children should be given the right to choose according to their level. Different applications should be researched and tried. There must be differentiation from formal education. Applications should be made to develop higher-order skills in science and arts centres. In addition, the competences of the teachers working in the support classes should be developed. The parallel education model should be used for the child to get to know himself/herself." (S13)

> "In my opinion, the main problem is not the education provided, but the assessment, evaluation and placement processes. On the one hand, we want to develop their creativity and thinking skills. On the other, we evaluate them with tests." (S13)

> "Students should be taught foreign languages other than English and more variety in foreign language should be offered and BİLSEM students should be provided with opportunities to go abroad to learn language." (S22)

> " . . . It was stated that all the activities that students wanted to participate in overlapped and they could not keep up with them." (S52)

In the category of problems arising from the institution ($f$ = 54), the problems of the intensity of the program and having to participate in too many activities ($f$ = 14), insufficient infrastructure and a lack of materials ($f$ = 10), a low number of tenured teachers ($f$ = 7), and transportation to BİLSEM ($f$ = 4) come to the fore. Below are some sample statements from the selected studies regarding the category of problems arising from the institution.

"It is seen that pre-adolescent children appear to have a high level of difficulty participating in too many activities." (S35)

" . . . My weekly schedule is very boring, even the parts I can enjoy are not fun because my schedule is very busy." (S29)

"It is very unlikely that we will do all the activities. The reason for this is the lack of physical facilities and the lack of time. We have a shortage of teaching materials; we have to manage at least two different activities from different branches in the same classroom . . . " (S27)

"There should be more technological training materials." (S16)

"When education starts every year, we, as BİLSEM, start education 2–3 weeks later because every year, new teachers are assigned to our Centre. It is 2–3 weeks late for them to start work. It also takes a long time for our incoming counsellors (teachers) to learn that we are different from the regular school and that they should prepare activities accordingly. Despite this, all our teachers who have come so far have been really diligent. Rather than being a teacher and student, they become like brothers and sisters to us." (S47)

"Our most important problem is the distance of the institution and the insufficient conditions in the building. Each branch must have a separate workshop, parents provide transportation to school under difficult conditions, there are problems while waiting for students, the institution also needs to be developed in terms of equipment. Our biggest problem is the school building and its location." (S27)

"Transportation support can be provided to students. Let the parents pay the fee again, but the service that can take the children from their school to BİLSEM is required." (S46)

"Sports events should be organized; venues for various sports, facilities, perhaps extensive sports halls should be built." (S22)

As can be seen from Table 3, teacher problems in BİLSEMs are divided into three categories: problems arising from students, problems arising from the program, and problems arising from the institution. In the category of problems arising from students ($f = 34$), the problems of the absence of students ($f = 12$), reluctance and low motivation in students ($f = 7$), and students being tired ($f = 4$) come to the fore. Below are some sample statements from the selected studies regarding teacher problems in the category of problems arising from students.

"There are students who are absent. In addition, there are students who come with the pressure of their families. We have a problem of absenteeism, especially among high school students. As the grade level increases in BİLSEM, absenteeism increases, as well. Students have problems fulfilling their responsibilities." (S14)

"The intensity of students' programs in formal education institutions, extra study hours in these programs overlapping with the study hours to be spent in the Bilsem program, the full-time education in our province cause absenteeism." (S64)

"Students are indifferent. They see themselves as knowing everything but some of them are highly inadequate. They are not interested in activities. Some come because of the pressure of the family." (S14)

"Students are tired. Students who come to BİLSEM after school are naturally tired because they have been studying for seven hours at school; this tiredness is both physical and mental. They come here hungry and they cannot eat anything until they go home." (S37)

"The pressure they experience due to high expectations from them reduces their efficiency in activities. They have problems in adaptation." (S40)

"Students can demand excessive attention. Children who do well and receive attention in their regular schools expect the same attention here." (S37)

In the category of problems arising from students in BİLSEMs ($f = 40$), the problems of the programs not being functional and adequate ($f = 15$), problems experienced in the measurement and evaluation process of the program ($f = 8$), and the programs not being up to date ($f = 6$) come to the fore. Below are some sample statements from the selected studies regarding the teacher problems in the category of problems arising from the program.

"The objectives do not seem very sufficient for gifted students. Although the objectives do not seem sufficient for the knowledge level of gifted students, they are placed in a way that can be taken to the upper level." (S12)

"No. If the program is implemented word for word, I don't think the objectives have been achieved. They can never learn scientific research skills." (S28)

"14.29% of the teachers expressed a negative opinion about the content of the BİLSEM exam. It is the opinions in this category that process-oriented evaluation should be made and creativity should be measured." (S33)

"Structural problems. Integration to formal education. Lack of measurement and evaluation." (S6)

"The program should be updated on the basis of standard objectives having international validity." (S12)

" . . . I definitely think that the subjects should be addressed in an interdisciplinary model. The same subject should be taught by three different branch teachers at the same time. In this way, their multidimensional thinking skills can be developed." (S11)

In the category of problems arising from the institution in BİLSEMs ($f = 139$), the problems of a shortage of materials, inadequate physical conditions and infrastructure ($f = 28$), insufficient professional development opportunities ($f = 16$), inadequacies in the process of diagnosing students ($f = 13$), too many students and crowded classrooms ($f = 12$), and working hours and time ($f = 10$) come to the fore. Below are some sample statements from the selected studies regarding teacher problems in the category of problems arising from the institution.

"We do not have enough materials. We do not have a designated classroom, we use any room we find empty. The conditions are not very comfortable for students." (S30)

"There is a lack of in-service training. Especially novice teachers do not know what to do. They have difficulty in adapting to the centre." (S64)

"The inventory used to diagnose gifted children is insufficient." (S58)

"At the end of a week of course training, students should be taken to music diagnosis, which can reveal their creativity more clearly. I think that the diagnosis alone is not enough, and it is necessary to make diagnoses that measure creativity, ability to conduct musical research and to take initiative and participation." (S27)

"Groups are crowded. Students constantly change groups for various excuses, which negatively affects group synergy." (S17)

"While working in formal education institutions, we experience difficulties in adaptation when we move to Science and Arts centres. When everyone goes to class (school), Bilsem teachers work at home, and on Saturdays and in evenings when everyone is at home, we work. This might create problems while we are planning our lives." (S14)

"Teachers should be relieved economically, they should not have the thought of how to bring the end of the month, the morale and motivation of teachers should be kept high so that the negativities are not reflected on their students." (S59)

"Teachers' personal rights are limited due to the lack of a regulation on science and arts centres." (S58)

"Teachers assigned to the institution from other schools are not efficient because they do not have knowledge of the institution. Since they are temporary and they know that they will leave, they do not contribute to the development of the institution and they see such tasks as extra burden. I think all the teachers working here should be permanent and tenured teachers." (S54)

"The lack of cooperation between BİLSEMs and the ministry's inability to provide the necessary coordination cause differences between BİLSEMs. Provincial and district directorates of national education do not pay due attention to BİLSEMs and they remain as forgotten institutions. While BİLSEMs should organize frequent trips, observations and activities, the lack of cooperation with institutions such as governorships and universities is another problem in BİLSEMs." (S37)

## 4. Discussion and Conclusions

In the current study, the problems experienced by teachers working in Science and Arts Centers and students attending these centers were examined through the analysis of studies focused on Science and Arts Centers. At the end of the study, the problems experienced by students and teachers were categorized and presented separately. First, the problems experienced by students were divided into three categories: problems arising from themselves and their environment, problems arising from the program, and problems arising from the institution. The problems were listed in each category. Then, the problems experienced by teachers were also divided into three categories: problems arising from students, problems arising from the program, and problems arising from the institution. The problems were also listed in each category.

In the category of student problems arising from themselves and their environment, the problems of being excluded from their circle of friends, too many expectations for them, and not being well understood and communication problems came to the fore. In the category of student problems arising from the program, the problems of insufficient training programs, problems in the process of measurement and evaluation, and no foreign language other than English came to the fore. Finally, in the category of student problems arising from the institution, the problems of the intensity of the activities and having to participate in too many activities, insufficient infrastructure and a lack of materials, and the low number of tenured teachers came to the fore.

In the study of Akbüber, Erdik, Güney, Çimşitoğlu, and Akbüber [33], the Special Talented Student Workshop, which was held with 168 students from 48 provinces of Turkey, was discussed. In the workshop, problems arising from the high expectations from the students, the high pace, the problems in friendship relations, and the lack of permanent teachers, which support the present research, were expressed. Again supporting our research, in the study conducted by Sarı and Öğülmüş [34] on the problems encountered in Science and Arts Centers, it was revealed that the students had a busy schedule and they had problems harmonizing with their peers. Epçaçan and Oral [35] also stated that students experience absenteeism at BİLSEMs due to exam anxiety and fatigue. In a study conducted in the USA, it was determined that the environment of compassion, support, and respect provided to gifted individuals is important for the development of these individuals' skills, competence, and peer relations [36]. In a study conducted in schools in two federal states of Germany, it was determined that gifted children establish better social relations and show more interest in school in environments adapted to their abilities and intelligence [37]. Buescher and Higham [38] stated that the expectations for gifted students are high and they become tired from their considerable efforts to meet these expectations. According to the findings of a study conducted in the USA, the parents of academically gifted children are completely focused on academic success, the perfectionist attitudes of the parents enable the children to set high goals, and the high expectations of the parents negatively affect the gifted children [39]. According to the results of a study conducted in Israel that gifted

students studying in special classes have lower academic self-esteem and higher anxiety about being evaluated by others than their normal peers, the "psychological support of gifted students" is of critical importance [40]. In a study conducted by Van-Tassel Baska [41], it was revealed that the curriculum of gifted students should consist of comprehensive concepts, themes, and problems and that interdisciplinary issues should be included in specific topics.

On the other hand, in the category of teacher problems arising from students, the problems of the absence of students, the reluctance and low motivation of students, and students being tired came to the fore. In the category of teacher problems arising from the program, the problems of the programs not being functional or adequate, problems experienced in the measurement and evaluation process of the program, and the programs not being up to date came to the fore. Finally, in the category of teacher problems arising from the institution, the problems of a shortage of materials, inadequate physical conditions and infrastructure, insufficient professional development opportunities, inadequacies in the process of diagnosing students, too many students and crowded classrooms, working hours and time, insufficient financial opportunities, problems in personal rights, and a lack of teachers, particularly tenured teachers, came to the fore.

In the study by Bozan and Savi Çakar [42], in which they revealed the problems of teachers working at BİLSEMs, teachers continuously state that the course hours in Science and Arts Centers are not suitable for both teachers and students. The most frequently mentioned problems by the teachers, from an educational sense, are the lack of equipment and materials in the center they work at, the insufficient physical equipment of the centers, and the content of the activities included in the framework programs of the Science and Arts Centers, which supports the present research. Again in support of this research, in Şenol's [43] master's thesis, in which the views of teachers on the education program of gifted students were obtained, it was stated that the majority of teachers faced problems with the physical environmental conditions of Science and Arts Centers. In a study conducted in two counties in the US state of Virginia, it was concluded that teachers who received training for mentally gifted children (85.5%) were more successful in recognizing gifted children than teachers who did not receive training (40.3%) [44]. In a study conducted with 212 primary school teachers from six regions of Finland, it was determined that nearly half of the teachers did not receive any training on gifted student education, although 82% of them were willing to receive training on this subject [45].

A total of one hundred and two problems for students ($f$ = 106) and two hundred and two problems for teachers ($f$ = 213) emerged in the studies, together with the recurring problems. Since the opening of the first Science and Art Center in Turkey in 1995, both the number of BİLSEMs and the number of students and teachers attending BİLSEMs have continued to increase. In fact, this increase has accelerated even more in recent years. In the news on the website of the Ministry of National Education, Minister Özer said that "While the number of BİLSEM was 183 in 81 provinces, we increased this number to 225 by the end of 2021. Our target in 2022 is to open 125 new BİLSEMs and increase the number to 350" [46]. With this numerical increase in BİLSEMs, it is seen that the problems are also increasing and waiting for a solution.

According to the formal education statistics of the Ministry of National Education published annually, the number of students attending formal education is 18085943, the number of teachers is 1112305, and the total number of schools is 67125 [47]. When it comes to BİLSEMs, it is stated in the news published on the website of the Ministry of National Education that these numbers are approximately 63,000 students, 2223 teachers, and a total of 225 BİLSEMs [48]. Based on this numerical information, when formal education is compared with Science and Arts Centers, it can be said that 0.35% of all the students in formal education attend BILSEMs, 0.20% of all the teachers in formal education work in BILSEMs, and 0.36% of all the schools in formal education are allocated to BILSEMs. As can be seen from these figures, all Science and Arts Centers do not constitute even one percent of formal education. In the Science and Arts Directive of the Ministry of National

Education, one of the aims of Science and Arts Centers is defined as: "To train individuals who adopt, protect and develop the national, moral, humanitarian, spiritual and cultural values of the country, have the power of free and scientific thinking and a broad world view and can contribute to the development of the country as constructive and creative individuals" [49]. Gifted students are seen as the future of countries, and it is evident that all investments in them will be added value for humanity in the future [50]. While it is good news that the numerical figures related to BİLSEMs increase with each year, it is worrying that, although they constitute a very small portion of formal education institutions, they have to deal with many problems every year. Of course, wherever there are people, it is inevitable that there will be problems. However, it may be possible to eliminate some problems (physical conditions, transportation, etc.) with small initiatives. It is believed that the problems put forward with a holistic perspective in this study will be taken into account and resolved by the authorities in solving the problems in BİLSEMs.

## 5. Suggestions

1.  In order for BİLSEMs to fulfill their functions, first of all, it is necessary to improve their physical equipment, eliminate the deficiencies of equipment and materials, and solve the problem of teacher shortages.
2.  The problems experienced by students and teachers in Science and Arts Centers should be taken into account and resolved by authorities.
3.  Different dimensions, other than students and teachers, can be addressed and studied regarding the problems experienced in Science and Arts Centers.
4.  More in-depth studies, such as meta-syntheses, can be carried out for the problems experienced in Science and Arts Centers.
5.  A needs analysis of the curricula followed in Science and Arts Centers can be conducted in relation to the problems that were presented in the current study.

**Author Contributions:** Conceptualization, H.Ş. and N.A.; methodology, E.K.; software, M.A.Ö.; validation, S.K., M.E. and R.G.; formal analysis, S.K.; investigation, H.Ş.; resources, M.A.Ö.; data curation, M.E.; writing—original draft preparation, N.A.; writing—review and editing, E.K.; visualization, R.G.; supervision, S.K.; project administration, H.Ş. All authors have read and agreed to the published version of the manuscript.

**Funding:** This research received no external funding.

**Institutional Review Board Statement:** This study was approved by the ethics committee of Akdeniz University Social and Human Sciences Scientific Research and Publication Ethics Committee with the decision dated 7 April 2022 and numbered 143.

**Informed Consent Statement:** Not applicable.

**Data Availability Statement:** Any required data will be provided with a request in writing via the email of the corresponding author listed due to the privacy of informants.

**Conflicts of Interest:** The authors declare no conflict of interest.

## Appendix A

**Table A1.** Numbers, names, types, and years of the studies focused on the problems experienced by students and teachers in BİLSEMs.

| Code | Name of the Study | Type | Author(s) | Year |
|------|-------------------|------|-----------|------|
| S1 | Investigation of the Problems of Music Unit Students Studying at Science and Art Centers (BİLSEM) | Book | S. Çiloğlu, D. B. Çevik Kılıç | 2022 |
| S2 | Opinions of Music Teachers Working in BİLSEMs on Instrument Education of Gifted Children | Master's Thesis | Ç. Karagün | 2022 |
| S3 | Factors Affecting the Success of Education Reform: Analysis of the BİLSEM Model in the Context of Policy Attributes Theory | Analysis | B. Yakut Özek | 2021 |

**Table A1.** *Cont.*

| Code | Name of the Study | Type | Author(s) | Year |
|---|---|---|---|---|
| S4 | Evaluation of the Support Education Program Prepared for Gifted Students | Article | M. Polat, İ. Polat | 2021 |
| S5 | BİLSEM History Teachers' Workshop Preparation Experiences | Article | O. Akhan, S. Altaş | 2021 |
| S6 | Opinions of BİLSEM History Teachers About the Working Conditions in BİLSEM | Article | O. Akhan, S. Altaş | 2021 |
| S7 | Teacher Opinions on Science and Arts Centres Social Studies Curriculum | Article | H. Bolat, F. Karakuş | 2021 |
| S8 | Gifted Children from the Perspective of Their Parents | Article | H. Demirkaya, O. Ünal, İ. Bozan | 2021 |
| S9 | Comparison of Class Levels According to Expert Opinions in the Diagnosis of Gifted Students | Article | A. Eker, H. Sarı | 2021 |
| S10 | Longitudinal Evaluation of the Education Given in Science and Arts Centres According to Family Views | Article | N. Büyüktokatlı, A. Kurnaz | 2021 |
| S11 | Criticism of the Education Provided in BİLSEMs from the Perspective of Administrators and Teachers | Article | M. Yılmaz, T. Yılmaz | 2021 |
| S12 | Analysis of Science and Arts Centres English Curriculum Based on Teachers' Views | Article | H. Torunoğlu, M. Ünal, E. Karabay | 2021 |
| S13 | Opinions of Faculty Members on the Use of Distance Education in the Education of Gifted Students | Article | M. Alpaslan | 2020 |
| S14 | Determination of the Problems Experienced by Science and Arts Centre Teachers and Suggestions for Solutions to These Problems | Article | İ. Bozan, F.S. Çakar | 2020 |
| S15 | A Riddle for Gifted Students: Being in More than One Educational Institution | Article | F. Bahçeci, U. Epçaçan | 2020 |
| S16 | Investigation of the Opinions of Gifted Students on Chemistry Lesson: Case of Erzurum BİLSEM | Article | T. Başar Daz, Z. Karagölge, İ. Ceyhun | 2020 |
| S17 | Turkish Education Problems For Gifted Students and Solution Suggestions | Article | C. Şahin | 2020 |
| S18 | Investigation of the Creative Writing Skill Levels of Gifted Students | Article | B. Özcan, H. Kontaş, M. Polat | 2020 |
| S19 | Evaluation of Intelligence Tests Used in BİLSEM Diagnostic Process According to the Opinions of Psychological Counsellors and BİLSEM Teachers | Article | A. Kurnaz, S. Gökdemir Ekici | 2020 |
| S20 | Views of Students, Teachers and Administrators on BİLSEM Education Programs in the Context of Values Education | Master's Thesis | S. Orman | 2020 |
| S21 | Examining the Problems Experienced by Mathematics Teachers Working in BİLSEMs in Mathematics Education | Master's Thesis | G. Su | 2020 |
| S22 | A Method Proposal for the Evaluation of the Problems of Gifted Students in Science and Arts Centres "Gifted Students Workshop" | Article | B. A. Akbürer, E. Erdik, H. Güney, G. G. Çimşitoğlu, C. Akbürer | 2019 |
| S23 | Opinions of Gifted Students on Teaching Practices in BİLSEMs | Article | U. Epçaçan, B. Oral | 2019 |
| S24 | Opinions of Gifted Students on the Scientific Conversations Held in the Science and Arts Centres | Article | D. Girgin, İ. Satmaz | 2019 |
| S25 | Evaluation of the Opinions of Teachers Working in the Education of Gifted Students on Preparing Individualized Education Programs, Implementing and Monitoring Them | Article | S.S. Ilik | 2019 |
| S26 | Turkish Education for Gifted Students in Science and Arts Centres: A Case Study | Doctoral Dissertation | O. Alevli | 2019 |
| S27 | The Efficiency of Science and Arts Centres in the Education of Gifted Students | Master's Thesis | G. Akın | 2019 |
| S28 | Investigation of Science and Arts Centres Support Education Program According to Classroom Teachers' Views | Master's Thesis | A. Ertürk | 2019 |
| S29 | A Study on the Weekly Program of Gifted and Talented Children Who Will Shape Our Future | Paper | A. Yaşar Pırtı, M. Taşçı | 2019 |
| S30 | Opinions of School Administrators and Teachers on the Education Provided in the Support Room for Gifted Students | Article | K. Pemik, F. Levent | 2019 |
| S31 | Opinions of Visual Arts Teachers Working in Science and Arts Centres on Arts Education of Gifted Students | Article | F. Levent, F. Kansu Çelik | 2019 |
| S32 | Evaluation of Education Programs of Science and Arts Centres Based on Student Views | Article | E. Kayışdağ, M. A. Melekoğlu | 2019 |
| S33 | Determination of the Attitudes and Opinions of Classroom Teachers on the Education of Gifted and Talented Students | Article | N. G. Kaya | 2019 |
| S34 | The Role of Science and Arts Centres in the Education of Gifted Students: Teacher and Parent Opinions | Master's Thesis | S. A. Sarıay | 2019 |
| S35 | Psychological Counselling and Guidance Needs of Gifted Students in Turkey | Article | F. Altun, H. Yazıcı | 2018 |
| S36 | Adapting the Problem Screening Inventory for Gifted Students to Turkish Culture | Article | F. Altun, H. Yazıcı | 2018 |
| S37 | Problems Encountered by Mathematics Teachers Working in Science and Arts Centres | Article | A. Çetin, A. Doğan | 2018 |
| S38 | The Opinions of Science and Arts Centre Administrators on the Challenges and Solutions: The Case of Istanbul | Master's Thesis | G. S. Çoban | 2018 |
| S39 | Examining Computational Thinking Skills of Gifted Students | Article | S. Kirmit, İ. Dönmez, H. E. Çataltaş | 2018 |
| S40 | Problems Experienced by Science and Arts Centre Teachers in the Education Process and Student Orientation | Article | G. Batdal Karaduman, A. Elgün Ceviz | 2018 |
| S41 | The Role of Science and Arts Centres in the Turkish Education System | Article | B. Kuyumcuoğlu | 2018 |
| S42 | The Effect of Biology Project Studies on the Scientific Attitudes of Gifted and Talented Students | Article | M. Özarslan | 2018 |

**Table A1.** *Cont.*

| Code | Name of the Study | Type | Author(s) | Year |
|------|-------------------|------|-----------|------|
| S43 | Investigation of the Opinions of Turkish Teachers Working in Science and Arts Centres on the Individualized Education Plan | Article | M. Ateş | 2017 |
| S44 | Differences and Problems in Turkish Lesson Practices in Science and Arts Centres Individualized Education Program | Article | B. Bağcı Ayrancı, F. Mete | 2017 |
| S45 | Evaluation of the Diagnosis Process of Gifted Students in Our Country According to the Opinions of Teachers, Parents and Students | Master's Thesis | S. Gökdemir | 2017 |
| S46 | The Problem of Absenteeism of Science and Arts Centre Students | Book | H. Demirtaş, A. Culha | 2017 |
| S47 | Student Opinions on the Sustainability of Gifted Education in Science and Arts Centres | Article | H. Atlı, R. Balay | 2016 |
| S48 | Opinions of Classroom Teachers Working with Gifted/Talented Students in Support Education Rooms on This Application | Article | H. S. Tortop, S. Dinçer | 2016 |
| S49 | In-Service Training Problem of Teachers Assigned to Science and Arts Centres | Article | İ. Satmaz, İ. E. Gencel | 2016 |
| S50 | The Difficulties Encountered by the Parents of Gifted Children Attending the Science and Arts Centre (BİLSEM) (Case of Sakarya Province) | Master's Thesis | Ş. Çamdeviren | 2014 |
| S51 | Science Education Programs Implemented in Science and Arts Centres from the Perspectives of Administrators, Teachers and Students | Article | B. B. Ülger, S. Uçar, İ. Özgür | 2014 |
| S52 | Examination of Science and Arts Centre Students' Views on These Institutions | Article | Ç. Çelik Şahin | 2014 |
| S53 | Investigation of Opinions on the Setting of Standards in Science and Arts Centres in Turkey | Article | M. S. Summak, Ç. Çelik Şahin | 2014 |
| S54 | Evaluation of the Problems Encountered in Science and Arts Centres (BİLSEMs) on the Basis of Teacher and Student Views | Article | H. Sarı, K. Öğülmüş | 2014 |
| S55 | Evaluation of Science and Arts Centres in their Twentieth Anniversary Based on Reports and Administrators' Views | Article | A. Kurnaz | 2014 |
| S56 | The Differences Parents See in their Students Going to the Science and Art Centre (BİLSEM) | Article | A. Alkan | 2013 |
| S57 | Evaluation of Interdisciplinary Teaching Activities in Visual Arts Education of Gifted Students (Case of Konya BİLSEM) | Doctoral Dissertation | M. A. Genç | 2013 |
| S58 | Science and Arts Centres: Current Situations, Problems and Solution Suggestions | Article | M. Özer Keskin, N. Keskin Samancı, S. Aydın | 2013 |
| S59 | Evaluation of the Opinions of Teachers and Administrators Working in the Science and Arts Centre (BİLSEM) on Professional Development and School Development | Article | T. Altun, S. Vural | 2012 |
| S60 | Teachers' Opinions on Gifted Education Programs (Case of BİLSEM) | Article | İ. Y. Kazu, C. Şenol | 2012 |
| S61 | Evaluation of the Education Programs of the Science and Art Centre with the Hammond Model | Doctoral Dissertation | Y. Eser | 2011 |
| S62 | Teachers' Opinions on Gifted Education Programs (Case of BİLSEM) | Master's Thesis | C. Şenol | 2011 |
| S63 | A Study on Science and Art Centres (BİLSEMs), a Model in the Education of the Gifted | Master's Thesis | H. Yıldız | 2010 |
| S64 | Evaluation of Science and Arts Centre Application | Master's Thesis | B. Sezginsoy | 2007 |

# Appendix B

**Table A2.** Methods of the studies focused on the problems experienced by students and teachers in BİLSEMs.

| Code | Model | Sample/Study Group | Data Collection Tool | Data Analysis |
|------|-------|--------------------|-----------------------|---------------|
| S1 | Descriptive | 10 students in BİLSEMs | Open-ended questions | Descriptive analysis |
| S2 | Case study | 10 BİLSEM music teachers | Semi-structured interview form | Content analysis |
| S3 | Case study | 5 administrators and 10 teachers working in BİLSEMs | Semi-structured interview form | Theoretical thematic analysis |
| S4 | Phenomenology | 7 teachers and 10 students in BİLSEMs | Semi-structured interview form, support program evaluation teacher questionnaire | Content analysis |
| S5 | Descriptive | 72 BİLSEM history teachers | Semi-structured, open-ended questions | Content analysis |
| S6 | Descriptive | 27 BİLSEM history teachers | Semi-structured, open-ended questions | Content analysis |
| S7 | Nested design | 77 BİLSEM social studies teachers | BİLSEM social studies curriculum evaluation questionnaire, semi-structured interview form | Descriptive statistics, content analysis |
| S8 | Case study | 25 students in BİLSEMs and their parents | Semi-structured interview form | Content analysis |
| S9 | Semi-structured interview technique | 30 experts working in Guidance Research Center | Semi-structured interview form | Content analysis |
| S10 | Survey | Parents of 78 students in BİLSEMs | Interview form | Content analysis |
| S11 | Case study | 5 administrators and 10 teachers working in BİLSEMs | Semi-structured, open-ended questions | Content analysis |

**Table A2.** *Cont.*

| Code | Model | Sample/Study Group | Data Collection Tool | Data Analysis |
|---|---|---|---|---|
| S12 | Case study | 12 English teachers | Interview | Content analysis |
| S13 | Case study | Faculty members of the Gifted Education Department | Semi-structured interview form | Content analysis |
| S14 | Case study | 30 BİLSEM teachers | Semi-structured interview form | Content analysis |
| S15 | Case study | 11 students in BİLSEMs | Semi-structured interview form | Content analysis |
| S16 | Case study | 23 students in BİLSEMs | Semi-structured interview form | Content analysis |
| S17 | Case study | 14 different Turkish teachers from 14 different BİLSEMs | Interview | Content analysis |
| S18 | Causal comparative | 85 students attending BİLSEMs | Creative writing activities | Descriptive statistics |
| S19 | Case study | 18 BİLSEM teachers and 19 counsellors | Interview | Descriptive statistics, content analysis |
| S20 | Phenomenology | 17 teachers, 19 students, and 5 administrators from BİLSEMs | Semi-structured interview form | Content analysis |
| S21 | Phenomenology | 13 mathematics teachers working in BİLSEMs | Interview | Descriptive analysis |
| S22 | Case study | 168 students attending BİLSEMs | Interview and document analysis | Descriptive analysis |
| S23 | Case study | 56 students in BİLSEMs | Semi-structured interview form | Content analysis |
| S24 | Phenomenology | 42 students in BİLSEMs | Semi-structured interview form | Content analysis |
| S25 | Case study | 22 BİLSEM teachers | Semi-structured interview form | Descriptive analysis |
| S26 | Case study | 19 teachers, 29 students, and 19 parents from BİLSEMs | Semi-structured interview form, semi-structured observation form, reflective student diaries and documents | Content analysis and descriptive analysis |
| S27 | System approach | 23 students, 25 teachers, and 30 parents from BİLSEMs | Semi-structured interview form, observation | Content analysis and descriptive analysis |
| S28 | Survey, case study | 158 BİLSEM classroom teachers | Support education program evaluation questionnaire, semi-structured interview form | Descriptive analysis, basic level analysis, and content analysis |
| S29 | Focus group interview technique | 20 students in BİLSEMs | Interview form | Descriptive analysis |
| S30 | Phenomenology | 20 administrators and 19 teachers working in primary and middle schools | Semi-structured interview | Content analysis |
| S31 | Case study | 12 visual arts teachers working in BİLSEMs | Interview | Content analysis |
| S32 | Survey | 600 students attending BİLSEMs | Scale | Descriptive analysis |
| S33 | Descriptive survey | 220 classroom teachers in Ankara | Personal information form, attitude scale | Descriptive analysis, content analysis |
| S34 | Content analysis | 32 teachers working in BİLSEMs and 71 parents | Personal information form, open-ended question form | Content analysis |
| S35 | Cross-sectional survey | Parents of 606 students attending BİLSEMs | Information form, problem screening inventory for gifted students (ÜY-PTE) | Descriptive statistics |
| S36 | Not specified | Parents of 242 students attending BİLSEMs | Information form, problem screening inventory for gifted students | Descriptive statistics |
| S37 | Phenomenology | 13 BİLSEM mathematics teachers | Written interview form | Content analysis |
| S38 | Nested single-case design | 8 BİLSEM principals | Interview form | Content analysis |
| S39 | Relational survey | 59 students attending BİLSEMs | Scale | Descriptive analysis |
| S40 | Case study | 16 teachers working in BİLSEMs | Interview | Content analysis |
| S41 | Semi-structured interview | Teachers, students, and parents in BİLSEMs located in Karşıyaka and Narlıdere | Interview | Content analysis |
| S42 | Qualitative and quantitative and experimental | 46 students in a biology program | Attitude scale, semi-structured interview | Descriptive analysis |
| S43 | Interview technique | 19 BİLSEM Turkish teachers | Structured interview form | Descriptive analysis |
| S44 | Document analysis, interview method | 10 BİLSEM Turkish and literature teachers | Semi-structured interview form | Content analysis |
| S45 | Case study | 228 teachers, students, and parents in BİLSEMs | Structured interview form | Descriptive analysis and content analysis |
| S46 | Case study | 2 administrators, 7 teachers, 7 students, and 8 parents in BİLSEMs | Semi-structured interview form | Content analysis |
| S47 | Interview technique | 22 students attending BİLSEMs | Semi-structured interview form | Content analysis |
| S48 | Case study | 15 classroom teachers working in BİLSEMs | Semi-structured interview | Content analysis |

**Table A2.** *Cont.*

| Code | Model | Sample/Study Group | Data Collection Tool | Data Analysis |
|------|-------|--------------------|--------------------|---------------|
| S49 | Phenomenology | 30 teachers working in BİLSEMs | Semi-structured interview | Content analysis |
| S50 | Content analysis technique | Parents of 108 students attending BİLSEMs | Open-ended questions | Content analysis |
| S51 | Case study | 6 teachers and 5 administrators working in BİLSEMs and 10 students | Interview | Content analysis |
| S52 | Qualitative | 4th-grade and 5th-grade students attending Adana and Mersin BİLSEMs | Open-ended question form | Content analysis |
| S53 | Qualitative | 3 academicians, 3 administrators, 3 teachers, and 3 parents | Interview | Content analysis |
| S54 | Semi-structured interview | Teachers and students invited to the 1st National Child Congress | Semi-structured interview | Content analysis |
| S55 | Case study | 32 BILSEM principals | Interview and document analysis | Descriptive analysis |
| S56 | Interview technique | Parents of 7 students attending BİLSEMs | Semi-structured interview form | Descriptive analysis |
| S57 | Single-group quasi-experimental design | 17 students attending BİLSEMs | Teacher as an observer interview form, student reflection form, student interview form, expert evaluation form | Content analysis and descriptive analysis |
| S58 | Survey | Administrators and students in the selected BİLSEMs | Questionnaire, semi-structured interview | Descriptive statistics |
| S59 | Phenomenography | 20 BİLSEM teachers | Semi-structured interview technique | Continuous comparison analysis method |
| S60 | Survey | 24 teachers working in BİLSEMs | Questionnaire | Descriptive analysis |
| S61 | Survey | BİLSEM teachers, administrators, students, and parents | Questionnaire | Descriptive statistics |
| S62 | Survey | 318 teachers working in BİLSEMs | Questionnaire | Descriptive statistics |
| S63 | Descriptive survey | 170 teachers working in BİLSEMs, 269 students, and 238 parents | Scale | Descriptive statistics |
| S64 | Survey | 25 teachers working in BİLSEMs | Scale | Descriptive statistics |

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
