# Peer review of "A Systematic Compilation of the Problems Encountered by Teachers and Students in Science and Arts Centers in Turkey"

_sustainability, doi:10.3390/su15032537_

Round 1

Reviewer 1 Report

Dear authors,

I appreciate the relevance of your study, but the manuscript falls short.

I would like to give you some inputs to improve the manuscript:

- In the title of manuscript specify where the study was carried out;

- the abstract needs to be restructured (improved). I suggest: one sentence providing a basic introduction to the field, comprehensible to an academic; one/two sentences of more detailed background; one sentence clearly stating the general problem being addressed by this particular study; one sentence summarizing the main result; one/two sentences explaining what the main result reveals in direct comparison to what was thought to be the case previously, or how the main result adds previous knowledge; finally one sentence to put the results into a more general context.

-I would suggest that tables 1 and 3 should be in Annex 

- Discussion of results needs framing/comparison with similar studies in other countries (ex: indexed publications).

-The statement "It is believed that the problems put forward with a holistic perspective in this study will shed light on the steps to be taken towards BİLSEMs" (lines 468-470) must be focused on the results (What are the steps?).

Rew

Author Response

Dear reviewer,

Thank you for your comments and editing suggestions. These recommendations are important to us. We made adjustments in line with these suggestions. Below are the details of these regulations. Thanks.

-Added where the work was done to the title of the article.

- Abstract edited. Added a basic introductory sentence. Purpose statement made clearer.

- Table 1 and Table 3 have been added to the Appendices.

- Studies from other countries have been added to the Results section. Compared with other countries.

-The confusion in the expression "It is believed that the problems put forward with a holistic perspective in this study will shed light on the steps to be taken towards BİLSEMs" has been fixed. It was rearranged.

- In addition, the article was sent to the English editing department of Sustainability and professional editing was done.

Reviewer 2 Report

The manuscript aims to analyse difficulties experienced by teachers and students  attending  Science and Arts Centres  in Turkey. It addresses issues relevant to special and inclusive education of gifted and talented students rather that sustainable education. The manuscript is presented in a well-structured manner, but language is not always clear, grammatically correct and academic.  The presentation of the problem, research and discussion form a logical sequence, leading to important indications for Turkish educational system, but international context is not represented enough in the discussion.    

The introduction should have referred to the categories of sustainable development, sustainable education, clearly indicating the need for an in-depth analysis of the problem in relation to these categories. The literature review refers to clarifying definitions and indicating the legal basis for the centers in Turkey. Solutions adopted in other countries are mentioned as well. The text should be coherent as a whole. It is advisable to refer to Renzulli's theory in the following part of the text (discussion and recommendations), since the classification of students for classes takes into account only their intellectual level and no other categories indicated in the theory, such as motivation and creativity. Reference should also be made to foreign experience in the discussion of the results. The text is unclear, ungrammatical and unacademic in places. Some sentences are unfinished, punctuation is used incorrectly, which hinders the clarity of the text (58-63, 70-73; 119-120), some terms are unacademic (“he”-85 instead of “he/she” as it as in the rest of the text; “normal” instead of “regular”-96; “church grove”-99; “privileged”-106). There is no need for full names of documents (107-110). The research problems are clearly defined, but the phrase "over the years" is unclear.

The choice of method is justified, the criteria for searching for data are specified. The research results are presented in detail, but the text should be shortened for clarity. I suggest that Tables 1 and 3 be included in the supplementary materials due to their volume. Table 2 can be replaced with an description.  Table 3 should be replaced with a summary table. Some clarification is needed on the term: S38 "Nested single case design" Is Nested single case design a Single-Case Designs (SCD) or A nested case-control (NCC) or a compilation of both? In Table 4, some  statements can be combined into more general categories, (e.g. such “Being shown as an example, even though they don't want to stand out”, and “Being seen as someone better at school”;  “Insufficient individualization and boredom at school” and “Academic conversations are boring and ineffective”.

It would be advisable to select a smaller number of examples of teachers' statements, choose the most represented ones, and avoid repetition.

In the discussion, it is advisable not to repeat previously presented results, but to discuss them with the studies conducted in other countries. It is also advisable to refer to inclusive education as a contemporary model for educating students with special educational needs.

In the bibliography, 10 items out of 40 were published in the last 5 years (suggested literature e.g.  Talent Development in Gifted Education Theory, Research, and Practice by Joyce VanTassel-Baska, 2022). 

Author Response

Dear reviewer,

Thank you for your comments and editing suggestions. These recommendations are important to us. We made adjustments in line with these suggestions. Below are the details of these regulations. Thanks.

- Sustainable development and sustainable education categories were referred to in the introduction. The 2nd and 3rd bibliography contain these regulations.

- In the introduction, Renzulli's theory and thoughts were referred 4 times.

- Foreign experience is also referred to in the discussion of the results. Comparisons with other countries are also included.

- The phrase "over the years" has been edited. Made understandable.

- Method selection justification and data search criteria are included in this section.

- Table 1 and Table 3 have been added to the Appendices.

- S38 "Nested single case design" such terms are taken as is from the included works.

- Fewer examples of teacher statements were included. 4 samples were removed.

- In Turkey, gifted students normally receive education together with non-gifted students. He receives gifted education outside of school hours. Inclusive education is applied to students with special needs.

- The bibliography section was arranged according to the template.

- In addition, the article was sent to the English editing department of Sustainability and professional editing was done.

Reviewer 3 Report

The article is written in accordance with the requirements of the journal, but I do not see the added value of the content for the general public. The analysis of the reviewed material is, in my opinion, suitable for internal use and own appreciation of the existing situation.

Author Response

Dear reviewer,

Thank you for your comments and editing suggestions. These recommendations are important to us. We made adjustments in line with these suggestions. Below are the details of these regulations. Thanks.

- Studies from other countries have been added to the Results section. Compared with other countries.

- In addition, the article was sent to the English editing department of Sustainability and professional editing was done.

Round 2

Reviewer 1 Report

Dear authors:

I have greatly appreciated the improvements. Your manuscript has now been made clearer and scientifically sound.

Rew

Reviewer 2 Report

The introduction and conclusion refer to the concept of sustainable development. The text is clear after proofreading and long tables being transferred to Appendix.  In the discussion, results were discussed with the studies conducted in other countries. New literature was added to bibliography.

Reviewer 3 Report

The authors have restructured the article and improved it even more, but I can see a significant contribution to the field.